# Transcriptomic Biomarker Signatures for Discrimination of Oral Cancer Surgical Margins

**DOI:** 10.3390/biom12030464

**Published:** 2022-03-17

**Authors:** Simon A. Fox, Michael Vacher, Camile S. Farah

**Affiliations:** 1Australian Centre for Oral Oncology Research & Education, P.O. Box 285, Nedlands, WA 6909, Australia; simon@oralmedpath.com.au; 2Australian eHealth Research Centre, The Commonwealth Scientific & Industrial Research Organisation, Floreat, WA 6104, Australia; michael.vacher@csiro.au

**Keywords:** oral squamous cell carcinoma, surgical margins, gene expression profiling, gene biomarker signature, multivariate statistics

## Abstract

Relapse after surgery for oral squamous cell carcinoma (OSCC) contributes significantly to morbidity, mortality and poor outcomes. The current histopathological diagnostic techniques are insufficiently sensitive for the detection of oral cancer and minimal residual disease in surgical margins. We used whole-transcriptome gene expression and small noncoding RNA profiles from tumour, close margin and distant margin biopsies from 18 patients undergoing surgical resection for OSCC. By applying multivariate regression algorithms (sPLS-DA) suitable for higher dimension data, we objectively identified biomarker signatures for tumour and marginal tissue zones. We were able to define molecular signatures that discriminated tumours from the marginal zones and between the close and distant margins. These signatures included genes not previously associated with OSCC, such as *MAMDC2*, *SYNPO2* and *ARMH4.* For discrimination of the normal and tumour sampling zones, we were able to derive an effective gene-based classifying model for molecular abnormality based on a panel of eight genes (*MMP1*, *MMP12*, *MYO1B*, *TNFRSF12A*, *WDR66*, *LAMC2*, *SLC16A1* and *PLAU*). We demonstrated the classification performance of these gene signatures in an independent validation dataset of OSCC tumour and marginal gene expression profiles. These biomarker signatures may contribute to the earlier detection of tumour cells and complement existing surgical and histopathological techniques used to determine clear surgical margins.

## 1. Introduction

Oral squamous cell carcinoma (OSCC) is the most common and deadly head and neck cancer (HNC), affecting more than 750,000 people worldwide [1]. In the last three decades, diagnostic progress in OSCC has been limited, and overall patient survival has improved by less than 10%, with 5-year survival rates of 8–27% reported for patients with metastatic disease [2,3]. Therapeutic protocols for OSCC include surgery, radiotherapy and chemotherapy; however, surgical resection of tumours remains the cornerstone of the initial definitive management of OSCC. The outcomes of surgical resection of OSCC have long been affected by limitations in the evaluation of surgical margins using pre-operative white light visual examination, palpation of the lesion and microscopic evaluation of frozen sections at a histopathological level. Regional failure and distant tumour recurrence continue to account for the high mortality rate of OSCC patients. Local relapse rates of 20% have been reported for oral cancer cases with confirmed tumour-free “non-involved” surgical margins defined by traditional examination [4,5]. This local relapse may occur due to minimal residual disease from incomplete resection that is not detected by routine histopathological investigation or premalignant tissue that gives rise to a later tumour through field cancerisation [6] or through independent multiclonal expansion of precursor tissue [7]. The ability to detect residual abnormality in oral cancer surgical margins using highly sensitive molecular techniques to assay for specific biomarkers would offer a superior assessment of surgical performance and inform decision-making regarding postoperative treatment. Once confirmed, these markers may be used for real-time in vivo endomicroscopic assessments of tumour margins, enabling live molecular imaging. Such interventions have the potential to guide the surgical removal of tumours and improve patient outcomes by minimising the retention of abnormal tissue within the surgical bed.

Many different statistical methods have been applied to the identification of biomarkers; however, generally, variables are evaluated for statistically significant behaviour between sample groups using classical statistical tests by focussing upon candidate biomarkers individually. When working with highly complex datasets such as those generated by transcriptome analysis, classification techniques can be extremely powerful in the identification of biomarkers [8]. In contrast to classical methods, multivariate methods incorporate the correlation structure of the data and accommodate interactions among candidate biomarkers. This approach has the potential to develop biomarker panels with superior prognostic performance to individual biomarkers with regards to sensitivity, specificity and reliability [8]. Partial least squares-discriminant analysis (PLS-DA) is a multivariate regression method that has been widely used for classification purposes and biomarker discovery in large datasets such as metabolomics [9], proteomics [10] and transcriptomics [11]. Sparse PLS-DA (sPLS-DA) is an extension of the technique that improves the application of classification and biomarker discovery by providing automatic variable selection [12].

Previously, we have reported that the margins determined using Narrow Band Imaging (NBI) harbour less molecular abnormality than those derived using conventional white light techniques [13,14]. In these studies, we have generated global mRNA and microRNA (miRNA) expression data for tumour (T), conventional margins determined by white light examination (M) and a more molecularly distant margin determined by NBI (N). We know from our own data [13] and from clinical evidence of local recurrence [5] that tumour-adjacent tissues may harbour significant aberrations. Here, we use the more distant NBI-defined surgical margins as the control tissue to build a predictive biomarker panel for molecular abnormality in the tumour margin. We apply sPLS-DA to develop classification models for the discrimination of tumour margins and the identification of molecular abnormality in the margins. More importantly, we demonstrate the generalisation of our biomarker statistical models by validation in an external dataset.

## 2. Materials and Methods

### 2.1. Patient Group and Sample Collection

The details of the patient group and sample collection have been previously described [13,15]. Briefly, eighteen patients with oral cavity SCC were enrolled prospectively. Patient demographics, tumour characteristics and surgery have been described by us previously [13,15]. The study conformed to the principles outlined in the Declaration of Helsinki (2008), with ethical approval from the Royal Brisbane and Women’s Hospital Human Research Ethics Committee (HREC/08/QRBW20 and HREC/10/QRBW336). All subjects provided written informed consent to participate in the study. Prior to surgery, the OSCC sites were visualised using conventional white light illumination and then Narrow Band Imaging (NBI) using an Olympus NBI ENF-VQ nasendoscope with CLV-180 light source and processor (Olympus Medical Systems Corp., Tokyo, Japan). The OSCC resection was taken to ≥5 mm beyond the NBI-defined surgical margin; following which, 4-mm punch biopsies were taken from: (1) 5 mm beyond the limit of tissue abnormality visible by NBI (defined here as normal (N)), 5 mm beyond the limit of tissue abnormality visible by conventional white light (defined here as the margin (M)) and the core of the primary tumour (defined here as T). The relative locations of the sampling sites have been illustrated by us previously (Figure 1 in Reference [13]). The tumour histology was predominantly moderate (10)-to-well (7)-differentiated OSCC, with one case of verrucous carcinoma. Independent histopathology indicated “clear” margins for all but one resection, although only two had margins with >5 mm separation from disease. One “close” margin had just 1 mm of separation. Applying this approach, five-year disease-free survival (DFS) was 84.21%, and the local recurrence rate (LRR) was 5.26% [15].

### 2.2. Gene Expression Profiling and Bioinformatics

The detailed protocols for gene and miRNA expression profiling have been previously described by us [13,14]. The processed transcriptome and miRNA expression matrices from these studies were used in the present study and are available at ArrayExpress: E-MTAB-5933 and E-MTAB-6470. Briefly, mRNA expression data was obtained using Human Genome U133 Plus 2.0 Arrays (Affymetrix, Santa Clara, CA, USA). Following quality control filtering, the remaining array data was pre-processed using the GCRMA normalisation method [16]. Individual mRNA probes were mapped to genes using the manufacturer’s annotation data. In cases where multiple probes were associated with a single gene, the probe with the highest Median Absolute Deviation was kept. Additionally, probes not located within a gene were discarded from the analysis. The functional annotation of the genes was performed using the DAVID [17] and Harmonizome [18] databases.

### 2.3. Sample Classification and Feature Selection

To identify the classifying gene (or miRNA) expression signatures characteristic of different tissue zones (N, M and T), we performed sPLS-DA [12], as implemented in the mixOmics R package [19]. This method combines sample classification and feature selection in a one-step process, allowing the identification of key variables (i.e., genes) specific to each group (tissue zone). To maximise the classification performance, we identified the optimal number of components and number of variables to select per component using the “*tune*” method implemented in the mixOmics package. Briefly the “*tune*” function uses a grid search approach to test multiple combinations of parameters (number of components and number of variables selected on each component) and identify the set of parameters providing the lowest error rate. Specifically, we used this function to perform a 10-fold cross-validation, repeated 10 times and with a range from 5 to 100 variables per component. The set of parameters providing the lowest Balanced Error Rate (BER) were retained to fit the final models. The performance and discriminative ability of the final classifier were assessed using the Area Under the Curve (AUC) approach, averaged over 10 cross-validations using one vs. all comparison. The plotting functions of the mixOmics package were used to generate all plots except the loading plots, which were generated from the mixOmics output using Prism v 6.07 (GraphPad Software).

### 2.4. External Validation Dataset

An external gene expression dataset was used for additional validation of the classifying signature biomarkers downloaded from the GEO database (https://www.ncbi.nlm.nih.gov/geo/ accessed on 11 June 2021): GSE31056 [20], from which data was obtained for oral cavity OSCC tumours, surgical margins and adjacent normal tissue for 23 patients from a Canadian population. This dataset was generated using the same Human Genome U133 Plus 2.0 Array platform used in our study.

## 3. Results

### 3.1. Differential Gene Expression Profiling

The results of the differential gene and miRNA expression analysis from the three pairwise comparisons between the N, M and T sample groups have been previously reported by us [13,14]. In brief, there were a total of 4794 genes differentially expressed with individual comparisons showing 4387 genes for T vs. N, 3266 genes for T vs. M and 7 genes for N vs. M. Similarly, there were a total of 119 miRNA differentially expressed with individual comparisons showing 109 miRNAs for T vs. N, 81 miRNAs for T vs. M and 7 miRNAs for N vs. M. Thus, as we have previously reported, the N samples were more molecularly distinct from the tumours than the M samples, supporting the use of an adjunctive optical imaging approach for surgical resection of the oral cavity SCC [15].

### 3.2. Identification of Classifying Biomarkers for Tumour, Margin and Normal Tissues

Initially, we explored whether we could derive classifying features in the gene expression data using sPLS-DA for multinominal discrimination across all three tissue locations to assess the classifying performance and identify genes that may be of biological significance. Following preliminary modelling using the dataset of differentially expressed genes and tuning to identify a classifying subset, we derived a set of genes with optimal performance for distinguishing the three groups. The classifying performance of this model is presented in Figure 1A,B based on the expression of 14 genes, which were independently weighted within the model (Figure 1C). Expression of these genes across all sample zones is illustrated in Figure 1D. Using this model, discrimination of the T samples was highly effective; however, discrimination of the normal (N) and margin (M) samples was less effective (Figure 1A,B). Therefore, we removed the T samples and repeated the analysis to derive a binomial classifier for the remaining samples and reveal molecular differences between the normal and peritumoral tissue. The resulting binomial model consisted of 20 genes (Figure 2C) with good classifying performance for discriminating N and M samples (Figure 2A,B). This analysis indicates that marginal tissue, while histologically tumour-free, can be readily distinguished from the more distant normal tissue in our dataset using this panel of 20 gene expression biomarkers.

In addition to the gene expression dataset, we also analysed the comprehensive microRNA expression data available for the same samples that we explored as another source of classifying features across the three tissue groups using sPLS-DA. Using a similar strategy, we identified differentially expressed miRNA by pairwise comparisons between the sample groups and derived a classifying model based upon miRNA expression alone. We found that the classification performance of the miRNA expression-based model (Figure 3A–C) was broadly inferior to the gene expression model, especially for the T and M zones (Figure 1A). We also examined the performance of elastic net regression [21] for the identification of biomarkers and sample classification using the gene expression dataset. When we ran this process for multinomial classification of the N, M and T samples using a training set consisting of two-thirds of the dataset, we were unable to identify a subset of genes optimal for building a prediction model, and we consistently found that elastic net regression gave an inferior discriminating performance to sPLS-DA. As a consequence, we focussed on the application of sPLS-DA and the use of mRNA-based signatures only.

### 3.3. Development of a Biomarker Signature for Classification of Tumour Margins

In order to develop discriminating biomarkers able to identify residual abnormality in OSCC surgical margins, our strategy was to first identify classifier genes that were upregulated in tumour tissue and could reliably be applied to distinguish tumour tissue from normal tissue. These biomarkers could then be used to interrogate the corresponding peritumoral margin (M) biopsy samples.

We used sPLS-DA to derive a classifying gene expression panel that discriminated T from N samples using the subset of genes that were significantly upregulated in tumour tissue. For this T vs. N analysis, cross-validation tuning of the parameters showed that the best discriminating performance using sPLS-DA occurred when eight genes were selected (Figure 4A). Following identification of the optimal values, the methods were applied to the entire dataset of 31 samples (18 T and 13 N), and a list of genes for binomial classification were obtained. For sPLS-DA, the eight genes of the model and their weightings are shown in Figure 4A, and as expected, their classification performance for T vs. N was excellent (Figure 4B). This gene signature sPLS-DA model was applied to the 18 marginal samples, where 13 samples were predicted as normal while five were predicted as tumours (Figure 4C). Some predictions were unambiguous (e.g., samples 53-18-M as T and 50-17-M as normal), while others were more borderline (e.g., samples 35-12-M or 29-10-M as T), as shown in the prediction score plot (Figure 4C). The expression of these classifiers in all samples across the entire dataset is illustrated in Figure 4D. This gene signature and the associated algorithms represent tools that can be applied toward the classification of tissues.

### 3.4. Validation in External Datasets

In order to test the generalisability of our model and investigate its clinical applicability, we performed validation using an independent OSCC dataset of tumour, marginal and normal samples [20]. Initially, we explored this dataset using the model trained with all three zones and all significant expressed genes as derived in Figure 1A. We applied this model to the validation dataset, with the resulting confusion matrix of the classifying performance shown in Figure 5A. For this dataset derived from a Canadian OSCC population, the multinomial three-way classifier performed well for the classification of tumour (T) samples but performed poorly for discrimination between normal and margin samples (Figure 3A). This may reflect difficulties in classifying these samples or differences between the sampling strategies between the studies, since our own study used NBI to define the N zone for sampling [13]. We next used our binomial classifying model for T vs. N, as shown in Figure 4A, and then applied it to all marginal (M) samples in the validation dataset. For this population, there was gene expression data for multiple marginal zone samples for most patients, and we applied our classifier to all these samples. Expression of the classifying gene panel in all the samples in the validation data (N, M and T) is visualised by a heatmap in Figure 5B. Of 49 M samples from 24 patients, 13 were classified as tumours (T) by our classifier (Figure 5C, prediction score 0.5 or greater). We next examined whether the presence of molecular abnormality in the margins as classified by our model may have value in the prediction of recurrence. We stratified the patients in the independent cohort, with at least one margin predicted as T in our analysis, into a “high-risk” group. We then performed a survival analysis, which showed that the differences between the groups were not statistically significant (Figure 5D). These results demonstrated that, while our classifying signatures were effective in identifying tumour samples in the independent cohort, the statistical model signature of abnormality in the margins could not be used to discriminate the samples associated with recurrence.

## 4. Discussion

Several studies have investigated transcriptional dysregulation and disease-specific signatures in OSCC in the context of investigating carcinogenic processes, prognostic signatures and molecular targets [22,23]. Inadequate surgical removal of the primary tumour is one important factor associated with patient survival [24]. The presence of residual disease or epithelial dysplasia in the surgical resection margins is recognised as a contributor to disease recurrence, and molecular analysis is one approach that has been explored to address this challenge [6,25].

In this study, we investigated a multivariate statistical method for its ability to identify classifying gene signatures for OSCC relative to normal tissue and for identifying biological features that distinguish OSCC margins. We also sought to examine the application of this methodology in the detection of abnormality in surgical margins. We were interested in exploring the application of sPLS-DA, since this relatively recently described statistical approach has demonstrated advantages in classification from high-dimensional data but has not as yet been widely applied to tumour classification and biomarker derivation [12]. Our strategy used paired tissue samples from the same patient group, since that molecular context should be more informative for the investigation of abnormality in surgical margins, rather than normal tissue from unrelated patients. The methodological approaches to the identification of classifying gene signatures in OSCC to date have predominantly employed differential gene expression and focussed upon genes that are overexpressed in tumour tissue using univariate statistical methods.

Since classifying biomarkers derived from discovery-based analyses are frequently significant contributors to disease processes [8], we initially applied sPLS-DA to identify features that discriminated between the three tissue groups. Interestingly, it was the downregulation of a panel of genes expressed in N and M zones that was most effective at classifying T samples (Figure 1C), suggesting that the dysregulation of particular tumour suppressors is a more common feature for classifying tumour tissue. Both *MAMDC2* and *SYNPO2* have been characterised as tumour suppressors in solid cancers, although not previously described in OSCC [26,27], and *ABCA8* has been reported downregulated in OSCC [28]. In the tumour zone, we found the downregulation of *ROR1,* a molecule that has been associated with tumour progression and proposed as a therapeutic target in a number of cancers other than HNSCC [29]. Our findings of lower expression in OSCC tumour tissue are consistent with another recent study [30] and suggest a different role in this cancer. Furthermore, there is evidence showing *ROR1* is expressed in a number of normal tissues, including oesophageal epithelium [31], indicating that targeted therapies may result in toxicity in normal tissues, including oral epithelia, according to our expression data. These classifiers showed excellent discrimination of the T samples but were less effective at discriminating N and M. We speculated that the significant molecular differences between T and the other zones influenced the derivation of the biomarkers, and hence, we performed an analysis of the N and M samples only. In this way, we sought to explore the molecular differences that might differentiate marginal tissue from more distant normal sites to gain insights into genes’ driving biology in peritumoral tissue. We were able to generate a panel of genes that discriminated these zones. It should be noted that such markers and molecular differences can reflect both the potential for residual tumour tissue within the margin and also the influence of soluble tumour-derived factors upon the marginal cells and their compositions. Many of the classifying genes for the classification of normal and marginal tissues have not previously been described with reference to OSCC but are dysregulated in other cancers. *ARMH4* (C14orf37, UT2) has been implicated as a tumour suppressor in myeloma and was found here downregulated in margins relative to the normal tissue [32]. *FGFBP2* is an effector of cytotoxic cell activity and potentially antitumour immunity [33], and the downregulation seen in the marginal tissue may reflect the suppression of antitumour immunity.

For discrimination of the normal and tumour sampling zones, we were able to derive an effective gene-based classifying model for molecular abnormality based on a panel of eight genes (*MMP1*, *MMP12*, *MYO1B*, *TNFRSF12A*, *WDR66*, *LAMC2*, *SLC16A1* and *PLAU*). Since we ultimately aimed for biomarkers that could be used to detect abnormality in marginal tissues, we focussed upon genes upregulated in the tumour tissue. Many of the genes identified in the resulting classifying biomarkers have been previously identified as dysregulated in cancer, including *MMP1* and *MMP12.* Matrix metalloproteinases like *MMP1* and *MMP12* are key enzymes involved in extracellular matrix degradation during tissue remodelling and are frequently reported as upregulated in cancer, associated with invasion and metastases and potential therapeutic targets [34]. In OSCC, *MMP1* has previously been reported as overexpressed in tumours relative to normal tissue [28,35] and proposed as a biomarker of disease recurrence [20,36]. *MMP12* has been reported as upregulated in tongue SCC [28]. The upregulation of *MYO1B* has been associated with lymph node metastases and cellular invasion in HNSCC [37].

The immune-related gene *TNFRSF12A* has been significantly associated with overall survival (OS) in HNSCC, with the high-protein expression of TNFRSF12A correlated with poor OS [38]. *TNFRSF12A* was also significantly correlated with activated CD8+ T cells, CD4+ T cells, follicular helper T cells and regulatory T cells, implying the possible involvement of this gene in immunoregulation and the development of HNSCC.

*PLAU* (plasminogen activator urokinase) mRNA has also been shown to be significantly elevated in HNSCC tumour samples over normal specimens, with evidence for *PLAU* as an independent indicator for HNSCC prognosis [39] associated with poorer clinical outcomes [40]. Furthermore, *PLAU* expression has been positively correlated with the presence of M1-type macrophages and negatively associated with CD4+ T cells, Treg cells and follicular helper T cells. Intriguingly, this finding may be explained by evidence that *PLAU* can regulate the expression of *TNFRSF12A* [40], previously implicated in immune regulation in HNSCC [38]. Collectively, these results suggest that *PLAU* may be an independent biomarker for predicting the outcomes of HNSCC patients and potentially contribute to tumour immunosuppression either in concert with or via *TNFRSF12A* [39].

The laminin C gene family is associated with the progression of diverse cancers through epithelial-mesenchymal transition (EMT). *LAMC2*, which encodes the laminin subtype LN-332 γ2 chain, is associated with poor prognosis, infiltrating immune cells and higher levels of LAMC2 protein expression, particularly in tumour-budding areas in HNSCC contributing to proliferation, migration, invasion and metastasis [41]. *LAMC2*, a target gene of miR-134, activates the EGFR/MAPK pathway but is also downregulated by miR-134, thereby inhibiting the migration and invasion of OSCC tumour stem cells by suppressing the PI3K-Akt signalling pathway [42]. Not only is *LAMC2* a good prognostic biomarker candidate, but it also shows utility as an early diagnostic marker for OSCC [43]. Previously, we examined miRNA-mRNA interactions in this data and reported 100 putative miRNA–mRNA interactions between 40 miRNA and 96 mRNA [14]. We did not find the dysregulation of miR-134 in our data, suggesting that some other mechanism is responsible for the overexpression of *LAMC2* that we see here.

*SLC16A1* encodes the monocarboxylate transporter MCT1, a member of a family involved in transporting lactate into and out of tumour cells, serving as a key mediator in maintaining tumour microenvironment homeostasis [44]. Recently, *SLC16A1* has been shown to be upregulated in HNSCC tumour tissues compared to adjacent normal marginal tissues, emphasising its potential oncogenic role in cancer progression likely through the NF-κB pathway [45]. Additionally, *SLC16A1* expression was closely associated with smoking and drinking behaviours in HNSCC patients. Interestingly, *MMP1*, *LAMC2* and *SLC16A1* have been shown to be significantly upregulated in HNSCC but downregulated by S100A8/A9 expression, contributing to increased proliferation, malignant transformation and disease progression in HNSCC [46].

There has been limited investigation of the role of *WDR66* (WD-repeat-containing protein 66) in HNSCC. WD-repeat proteins are frequently upregulated in various cancers, impacting upon transcription regulation, cell cycle control and apoptosis. *WDR66* was reported as upregulated in oesophageal SCC, associated with poor overall patient survival and a driver of EMT, identifying *WDR66* as a novel marker for risk stratification [47]. In salivary adenoid cystic carcinoma, the knockdown of *WDR66* decreases cellular proliferation, migration and invasion [48].

The key test of such a model is its applicability across independent populations, and we selected datasets from international populations to test the generalisability of this model. While there are a number of studies that have reported gene expression data for tumour and matched normal tissues, the sampling sites for normal tissues are not always well-defined, and studies that have reported tumour, peritumoral margins and more distant normal tissues are limited. We were unable to find a comparable study to our own, which used NBI to define the sampling zones; however, we used an independent cohort of Canadian OSCC patients [20] with three similar sampling zones to benchmark our classifying models. The authors of this study focussed their investigation upon gene expression in marginal tissues associated with recurrence. Although there is some ambiguity in the definition of sample zone locations, we found that our three-zone classifying model was highly accurate in the prediction of T samples but classified all N and M samples as marginal. This may reflect differences in the sampling of N tissues between the studies given the use of NBI in our study or the relatively inferior prediction accuracy for N and M samples that we found in our own dataset using this model. We speculated that the classification of marginal samples using our binary classifier of (T vs. N) might have application in the prediction of recurrence. In our own primary dataset, such an analysis was not possible, because recurrence was extremely low due to the application of NBI for margin determination [15]. In the external dataset for which recurrence data was available, we found that the margin classifying tumour signature was not suitable for the prediction of recurrence in this cohort. It seems possible that the identification of tumour signatures in marginal tissue is insufficiently discriminating for the prediction of recurrence, and there may be distinct marginal gene expression signatures predictive of this. Moving forward, refinement of these biomarker panels will require testing in additional cohorts with the well-defined sampling of marginal tissue and populations with different genetic backgrounds. Furthermore, in some Asian populations, there are alternative risk factors such as betel nut chewing that have different carcinogenic mechanisms and that may impact the relevance of some biomarkers. As much as possible, these populations need to be included early in follow-up studies to ensure the broad applicability of biomarker panels in clinical practice.

In conclusion, we investigated the performance of multivariate statistical methods in the discovery of diagnostic biomarker panels for OSCC from transcriptomic data. The novel classifying gene expression signatures derived here will contribute to the ongoing development of diagnostic and prognostic biomarker panels for OSCC. In addition, many of the classifying genes have not been previously reported for OSCC and represent promising putative targets for further molecular analysis.

## Figures and Tables

**Figure 1 biomolecules-12-00464-f001:**
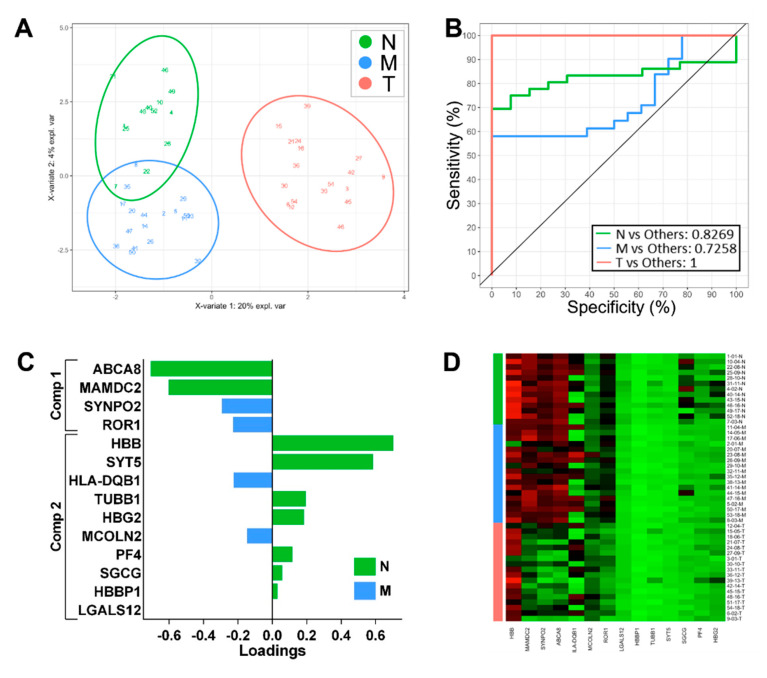
Classification performance sPLS-DA-derived gene expression signatures on all sample groups: N vs. M vs. T. (**A**) Two-dimensional sample plots for the multinomial classifier for 3 groups with confidence ellipse plots. (**B**) Corresponding ROC curves. (**C**) Loading plots for multinomial N vs. M vs. T classifiers showing the contribution of each gene to the signature. (**D**) Heatmap of the gene expression across all samples for the multinomial N vs. M vs. T classifiers.

**Figure 2 biomolecules-12-00464-f002:**
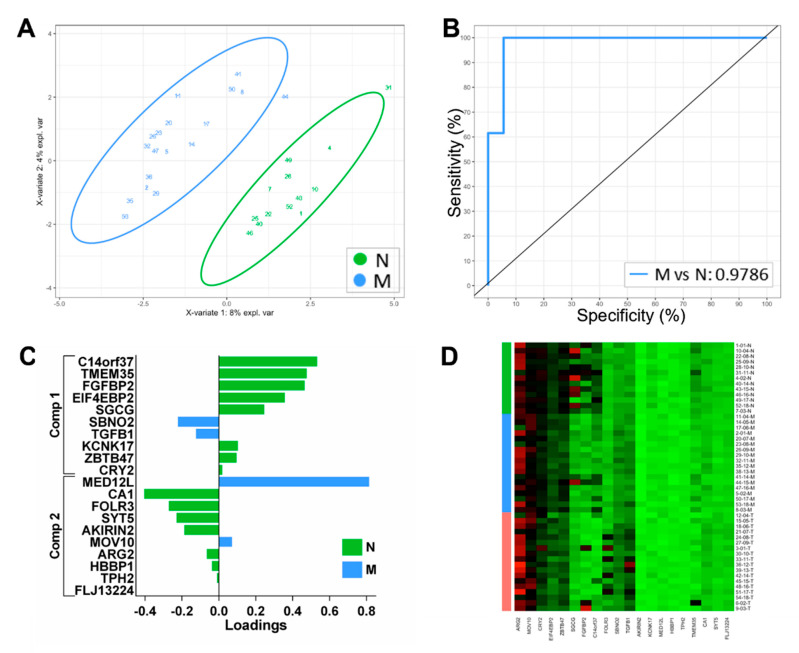
Classification performance sPLS-DA-derived gene expression signatures for binomial comparison: N vs. M. (**A**) Two-dimensional sample plots for the 2 groups with confidence ellipse plots. (**B**) Corresponding ROC curve. (**C**) Loading plots for N vs. M classifiers showing the contribution of each gene to the signature. (**D**) Heatmap of the gene expression across all samples for the N vs. M classifiers.

**Figure 3 biomolecules-12-00464-f003:**
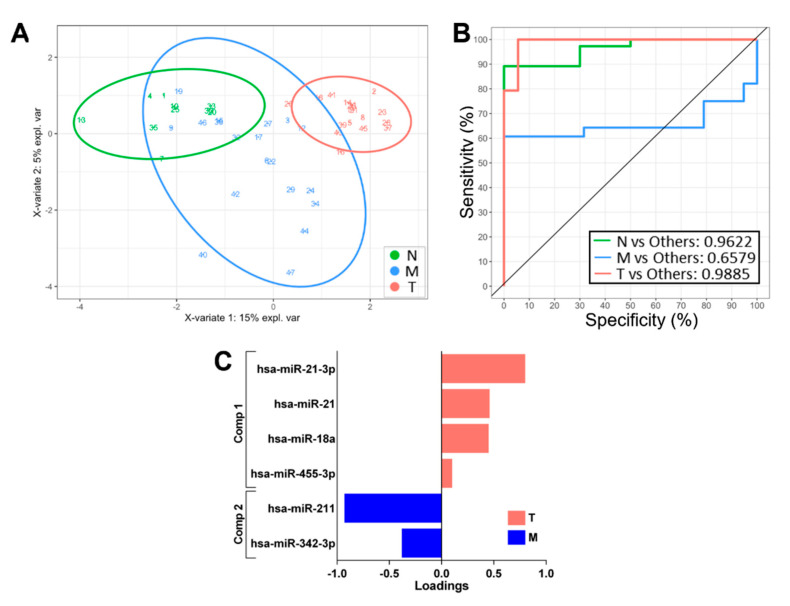
Classification performance of a sPLS-DA-derived miRNA expression signature. (**A**) Two-dimensional sample plots for the 3 groups with confidence ellipse plots. (**B**) Corresponding ROC curve. (**C**) Loading plots for multinomial N vs. M vs. T classifiers showing the contribution of each miRNA to the signature.

**Figure 4 biomolecules-12-00464-f004:**
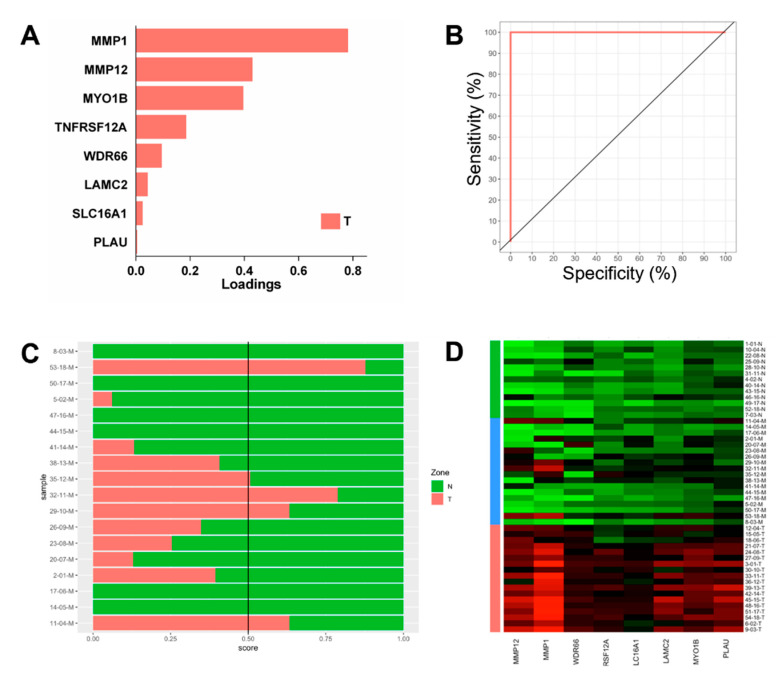
Classification performance of sparse Partial Least-Squares Discriminant Analysis-derived gene signatures for tumour vs. normal (T vs. N) using genes upregulated in tumours. (**A**) Loading plot of a discriminating gene panel. (**B**) ROC curve for tumour vs. normal discrimination. (**C**) Prediction score plot for the prediction of margins classed as T or N. (**D**) Heatmap of gene expression for the T vs. N gene models in all the samples.

**Figure 5 biomolecules-12-00464-f005:**
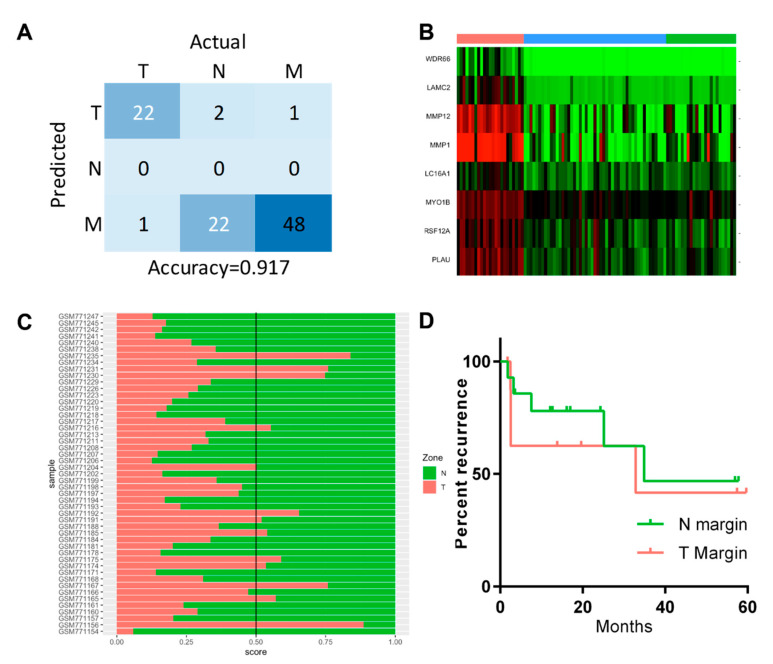
Testing sPLS-DA models in the validation data. (**A**) Confusion matrix of 3 zone predictions of the validation dataset using the classifiers from Figure 1. (**B**) Heatmap of the expression of T vs. N prediction model classifiers in the validation dataset. (**C**) Prediction score plot of the validation margins using the binary classifier for T vs. N in the validation data. (**D**) Survival plot for tumour recurrence for patients with primary margins predicted as T or N.

## Data Availability

Microarray data raw files and normalised expression data are available on ArrayExpress EMBL-EBI under database accession numbers E-MTAB-5933 and E-MTAB-6470.

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
