# Peer review of "Transcriptomic Biomarker Signatures for Discrimination of Oral Cancer Surgical Margins"

_biomolecules, 2022, doi:10.3390/biom12030464_

Round 1

Reviewer 1 Report

This was a study of gene expressions in oral cancer. The study included 18 oral cancer patients, possible 49 samples (tumor, margin by white light, margin by NBI in each patient; 5 NBI sample was excluded due to failed quality control?). The authors analyzed the gene expression obtained form their previous studies using partial least squares-discriminant analysis (PLS-DA). The cohort included 18 patients, 17 OSCC and one verrucous carcinoma. Nine of these patients were classified as stage I or stage II and others were stage IV.

The results showed the marginal tissue can be distinguished from the more distant normal tissue by 20 gene expression biomarkers. However, the classification performance of the miRNA model was inferior. The authors also found 8 genes significantly upregulated in tumor compared to normal tissue by sparse PLS-DA method. The external validation of this classification model by the GEO database showed 13 of 49 marginal tissue was classified as positive margin. However, the positive margin cases did not show a higher recurrence rate by Kaplan-Meier analysis.

Strength:

This study included 4794 genes for analysis. The introduction and discussion were well written.

Weakness:

A small sample size.

Although the results showed the classification model could identify molecular abnormality in surgical margins, these findings were not proved by clinical data.

Concerns:

Major concerns

  1. The statistical methods were not mentioned in the article. How sparse PLS-DA method was performed in this study should have more detailed descriptions.
  2. One of the patients was diagnosed as verrucous carcinoma. Verrucous carcinoma was histological difference with distinct tumor behavior compared to SCC. The author should consider to exclude this patient since the whole article was discussed about OSCC.
  3. In the section of 3.3. Development of a biomarker signature for classification of tumour margins.

The author found 13 samples were predicted as normal and 5 samples were predicted as tumor by their predictive model. I am curious if the 5 samples correlated to a less surgical margin.

Minor concerns:

  1. In line 95, a figure may help the readers to know where the samples were obtained.
  2. In line 102, the reference 15 included 20 patients but the present study included only 18 patients. The five-year DFS and local LRR should be modified for these 18 patients.
  3. In line 230, the figure 2A may not represent binomial classifying model for T vs N? (possible figure 4?)
  4. In line 359, this predictive model may not be generalized to some Asian countries with high prevalence of OSCC. Betel-nut chewing was one of the major risk factors in these countries and the gene expression may be significant difference. This condition may need to describe in discussion.

Reviewer 2 Report

In this study, the authors examined the transcriptomic levels of mRNA in discriminating OSCC from the tumor, close margin, and distant margin biopsies using sPLS-DA. They identify the diagnostic signature for earlier detection of the tumor cells that helps to define clear surgical margins. The novelty of the present study is acceptable.

Major points of improvement:

  1. Although the microRNA profile was hard to be used for building a prediction model to define surgical margin, the miRNA-mRNA network may be used to confirm the mRNA signatures. The authors may apply miRNAs’ target prediction webs, such as TargetScan or miRTarBase to predict the signature-related miRNAs to support the correlation of miRNA and the target mRNA. For example, the miR-134 and LAMC2.
  2. The authors should strengthen or confirm their findings of the molecular signature with clinical presentation. For example, the analysis of differential expression of the genes between tumor vs. normal tissues and build ROC curves by using TCGA-HNSC dataset.
  3. For each defined molecule, the authors should provide the value of sensitivity, specificity, and 95% confidence interval in the ROC curves and estimate the odds ratios to predict cancer risk.
  4. In Figure 3, the authors should identify and conclude the miRNA expression signature based on their microarray data. That is what miRNAs are significant?

Minor points of improvement: 

  1. Lines 225-227, Figure 3A showed the classification performance of sPLS-Da derived miRNA expression signature, but not mRNA signature. Please confirm it.
  2. Authors stated (lines 230-231) “…, We next used our binominal classify model for T vs. N shown in Figure 2A……… However, Figure 2A showed N vs M model. Please confirm it.

Round 2

Reviewer 1 Report

The authors had addressed my concerns and revised the article adequately. I thought this article is acceptable.

Reviewer 2 Report

This revision has significantly improved that showed scientifically valid.